# Development of a Simplified Portion Size Selection Task

**DOI:** 10.3390/foods10051121

**Published:** 2021-05-18

**Authors:** Aimee E. Pink, Bobby K. Cheon

**Affiliations:** 1School of Social Sciences, Nanyang Technological University, Singapore 639818, Singapore; aimee.pink@ntu.edu.sg; 2Department of Psychology, Swansea University, Swansea SA2 8PP, UK; 3Singapore Institute for Clinical Sciences, A*STAR, Singapore 117599, Singapore

**Keywords:** portion size, portion selection task, food characteristics, liking, cognitive restraint, online research

## Abstract

Portion size is an important determinant of energy intake and the development of easy to use and valid tools for measuring portion size are required. Standard measures, such as ad libitum designs and currently available computerized portion selection tasks (PSTs), have several limitations including only being able to capture responses to a limited number of foods, requiring participants’ physical presence and logistical/technical demands. The objective of the current study was to develop and test robust and valid measures of portion size that can be readily prepared by researchers and be reliably utilized for remote online data collection. We developed and tested two simplified PSTs that could be utilized online: (1) portion size images presented simultaneously along a horizontal continuum slider and (2) multiple-choice images presented vertically. One hundred and fifty participants (*M* = 21.35 years old) completed both simplified PSTs, a standard computerized PST and a series of questionnaires of variables associated with portion size (e.g., hunger, food item characteristics, Three Factor Eating Questionnaire). We found average liking of foods was a significant predictor of all three tasks and cognitive restraint also predicted the two simplified PSTs. We also found significant agreement between the standard PST and estimated portion sizes derived from the simplified PSTs when accounting for average liking. Overall, we show that simplified versions of the standard PST can be used online as an analogue of estimating ideal portion size.

## 1. Introduction

Portion size is an important aspect of eating behavior and self-selected portion size significantly predicts food intake [1] and plate clearing [2,3]. Consequently, a large division of eating behavior research has been dedicated to understanding the determinants of portion size [4,5,6] and targeting portion size within interventions to reduce obesity (for review see [7,8]). The development of useful, easy to use and valid tools for measuring portion size are required to further our knowledge of portion size.

The ‘gold standard’ measurement of portion selection behavior is analyzing how much one self-serves and/or consumes ad libitum [9,10,11,12]. However, this option is not always logistically or practically feasible to researchers. Such designs only have the capacity to measure single (limited amounts of) test foods and to test a single participant (small groups of participants) at a time. Furthermore, these designs require suitable laboratory space and equipment (e.g., a kitchen/dining area) which can be costly, in addition to the potential waste involved in preparing large quantities of standardized test foods [9].

Methodological advancements that address these potential constraints are computerized or simulated portion selection tasks (PSTs; standard computerized PSTs are referred to as/SCPST throughout) which have been commonly used in research on portion selection, premeal planning and expected satiation/satiety (e.g., [13,14,15,16,17,18,19,20,21,22,23]). In these tasks, participants can dynamically and flexibly change the portion of food presented on a computer screen using a keyboard. These tasks can be presented to large groups of participants simultaneously, depict a wide range of food items, reduce researcher burden on preparing food items and minimize the amount of food waste generated.

The SCPST has been shown to be a valid proxy of participants’ self-served portions of real food [1,24,25]. Indeed, studies found portions selected digitally using a standard SCPST were significantly and positively correlated with actual self-selected portions and food intake, indicating good ecological validity [1,25], demonstrating that portion sizes selected within the SCPST can be used as an analogue for actual behavior.

That said, the SCPST has its own practical limitations. Due to the large number of images of precisely weighed food portions required in the task (e.g., some tasks use 50 images depicting portions increasing in 20 kcal portions from 20 kcal to 1000 kcal), it can be cumbersome to develop the stimuli. In addition, specialized programming and software are required to run the task, which requires dynamically and incrementally changing the image of the portion size depicted on the screen in response to corresponding key presses. Image and software requirements can also raise issues with transferring such tasks to online formats. Difficulties and disruptions can arise with the time required for images to load, the feasibility of embedding software into common survey platforms such as Qualtrics, and compromise of image size or loss of the ‘animated’ format depicted in the original task [26]. Moreover, there is an increasing trend to conduct psychological and behavioral research online [27,28] and this has been further exacerbated due to COVID-19 restrictions [29]. Consequently, a measure of portion size that can be readily prepared and administered online is needed.

The aim of the current study was to test the agreement of two simplified PSTs, which were tailored for online studies, with the SCPST. To ensure the developed simplified PSTs were predictively valid, they needed to replicate the relationships already established within the portion size literature (see Appendix B for overview). There is initial support for such relationships as shown by Spence and colleagues [30]. They utilized just six images depicting different portion sizes and found sex and uncontrolled eating to be the strongest socio-demographic and psychological determinants of portion size, respectively. They also found a significant association with cognitive restraint and liking, but found no relationship with body mass index (BMI).

However, a limitation of the study conducted by Spence et al. [30] was the limited number of foods presented in their portion selection task. In our study, we present a wide range of foods and a greater range of portion sizes. Our first simplified PST consisted of five images presented simultaneously along a horizontal continuum whereby participants were instructed to move a slider to indicate their ideal portion (similar to an analogue scale; SPST). This task gave participants the opportunity to select portions in between the presented images which provided a wider range of possible portions replicating to some extent the small increases/decreases of the SCPST. The second simplified PST consisted of seven multiple-choice images presented vertically, which participants could scroll down (up) the page to view progressively larger (smaller) portion sizes (MPST). Participants selected which images best represented their ideal portion and this task maintained the presentation format of the SCPST by allowing participants to scroll through portion sizes. The simplified PSTs formats were selected because they each captured aspects of the original SCPST and could be suitably incorporated into common survey platforms (e.g., Qualtrics).

The study was exploratory and therefore, no a priori predictions were made about which simplified PST would have similar psychometric properties to, or best agreement with, the SCPST. Assessing agreement between two measures allows researchers to examine whether a new method differs in a meaningful way or if it could be used interchangeably. We compared the different PSTs across a diverse range of food types, with participants selecting their ideal portion sizes for 25 common foods that varied in characteristics (e.g., snack vs. meal-type foods, amorphous vs. individual units). In order to determine predictive validity, we measured constructs known to be associated with portion size, such as current hunger, prior engagement with exercise and eating behaviors (i.e., cognitive restraint, emotional eating and uncontrolled eating), as well as food-specific characteristics (i.e., how liked, familiar and filling each food was) and tested how well these constructs correlated with the different PSTs. Full information and pre-registration can be found on the Open Science Framework website for this study (https://osf.io/vhzd2 created on 31 August 2020).

## 2. Materials and Methods

### 2.1. Participants

One hundred and sixty participants were recruited from a Singaporean university’s research participation system and received either SGD $5 or 1 research participation credit for completing the study. Participants can sign up to these platforms to take part in university wide research studies for payment or to fulfill course requirements. Inclusion criteria involved being between the age of 18–65 years (parental consent was obtained for participants under the age of 21 (based on regulations for consent in Singapore)) and no current or historical diagnosis of an eating disorder. Six participants were identified as outliers or had unusual responses (one participant had extreme values for all three PSTs, one participant had extreme scores on two PSTs (MPST and SCPST), one participant had an extreme value for age and food insecurity and three participants had difference scores between the SCPST and the SPST/MPST over +/-3SD (e.g., SPST = 453.44, MPST = 428.00, SCPST = 69.90). One participant completed half of the questionnaires and three did not complete the SCPST. A sample of 150 remained (females = 62.97%, *n* = 94, one participant opted not to disclose; age: *M* = 21.35 years old (*SD* = 2.07, range = 18–30 years). Singaporean was the most commonly reported nationality (74.7%, *n* = 112) and other nationalities included: Chinese, Malaysian, Indian, Indonesian, Vietnamese and British. The majority reported their ethnicity as Chinese (88.0%, *n* = 32) and other ethnicities included: Malay: 2% (*n* = 3); Indian: 7.3% (*n* = 11%); Eurasian: 0.7% (*n* = 1); and other 2.0% (*n* = 3, e.g., Caucasian, Vietnamese). One participant reported being married or in a civil partnership with the remainder of the participants reporting being single. All participants were currently enrolled university students. Average reported household income was SGD $11,388.24 per month (SD = 30,634.99, range = SGD $0–300,000). Whilst participants were asked to state their household income, it is possible some reported their personal income and as participants were university students, they may have reported SGD 0.

### 2.2. Measures

#### 2.2.1. Current Hunger

Participants were asked to indicate on a 100-point visual analogue scale (VAS; slider position started in the center) their current hunger level, fullness level, desire to eat and how much they feel they could eat based on the hunger assessment [31]. A composite current hunger score was calculated (fullness item reversed, α = 0.91).

#### 2.2.2. Exercise

Participants were asked if they had exercised prior to the study and if they indicated they had they were further asked how strenuous their exercise was (1 = *not at all* to 4 = *high*).

#### 2.2.3. Portion Size Selection Task (PST)

Participants completed three versions of the PST containing 25 foods commonly available in Singapore (e.g., chicken with fried rice, mixed salad, chicken nuggets, fruit salad and chocolate cake). Before each task, participants were presented with the same instructions which asked them to select their *ideal* or *perfect* portion size if they were to serve themselves the particular food item for their next meal and no other foods were available. Food items were randomly presented to participants in all three tasks. A full list of foods presented can be found in the Appendix A.

##### Standard Computerized Portion Selection Task (SCPST)

A modified version of the SCPST reported by Sim et al. [25] (adapted from Wilkinson et al. [1]) was used. Food items were presented across 50 images which depicted portion sizes ranging from 20 kcal to 1000 kcal in 20 kcal increments. Participants were presented with a large high-resolution image on the screen and used the right/left arrow keys to increase/decrease portions respectively (see Figure 1). The starting portion size was randomly selected for the 25 foods and varied across participants. Participants were asked to view all images before selecting the portion they would serve themselves for their next meal. Average portion size in kcal was calculated (α = 0.91) and showed good reliability.

##### Simplified ‘Slider’ Portion Selection Task (SPST)

Five images were taken from the SCPST which depicted portions sizes ranging from 20 kcal to 740 kcal in 180 kcal increments (Table 1). Increments were chosen so that clear differences in portion size were visible across the images. The images were presented along a slider (marker was centered to start) and participants were asked to move the marker to indicate their ideal portion size (see Figure 2). Participants were informed that moving the marker to the furthest left or right would indicate they would serve themselves the smallest and largest portions, respectively. Placing the marker between two images indicated they would serve themselves a portion in between the two images. The slider was scored 0–100 and scores were converted to calories using: (score × 7.2) + 20, e.g., 0 = 20 kcals, 100 = 740 kcals. An average was taken across all 25 foods (α = 0.94).

##### Simplified ‘Multiple-Choice’ Portion Selection Task (MPST)

Seven images were taken from the SCPST which depicted portion sizes ranging from 20 kcal to 740 kcal in 120 kcal increments (see Table 1). The images were presented vertically, and participants were asked to scroll through all images before selecting the portion they would serve themselves for their next meal (see Figure 3). Images were assigned their calorie content (e.g., image 1 = 20 kcal) and an average was taken across all 25 foods (α = 0.94).

#### 2.2.4. Food Characteristics

Participants were presented with a single image (380 kcal) of each food item and asked to rate how much they liked it, how filling and how familiar each food was on a sliding scale scored 0–100 (*not at all* to *very* much; slider position started in the center, randomized order). Average overall ratings of liking (α = 0.84), filling (α = 0.87) and familiarity (α = 0.91) were generated.

#### 2.2.5. User Feedback

Participants were asked to provide feedback on the two simplified PSTs. Feedback questions asked participants about the clarity and size of the images and how noticeable the differences in portion sizes were on a sliding scale (0 = *strongly disagree*, 50 = *neutral* (starting point), 100 = *strongly agree*). Participants were also asked (1) their preference between the two SPSTs in general, (2) which of them was the easiest method to use and (3) which was the easiest method to distinguish between portion sizes (rated as SPST, MPST or no preference). An open comment box was also presented to provide participants the opportunity to give additional feedback (see Appendix A section 4.0).

#### 2.2.6. Three Factor Eating Questionnaire R21 (TFEQ)

The TFEQ is a self-report measure of cognitive restraint (CR), emotional eating (EE) and uncontrolled eating (UE) [32]. Participants responded to 21 statements on a 4-point Likert type scale (1 = *definitely true to 4 = definitely false*) and an average score was calculated for each subscale. Higher scores indicate higher levels of each eating behavior and good internal consistency was found (CR: α = 0.77; EE: α = 0.93; UE: α = 0.81). The TFEQ is a widely used measure of self-reported eating behavior and all three subscales have sufficient reliability and validity [32,33,34].

#### 2.2.7. Food Insecurity

An adapted version of the US Department of Agriculture Food Security survey captured food insecurity [33]. Participants were asked to indicate how often in the past 12 months they experienced situations such as not being able to afford balanced meals and feeling hungry but being unable to eat as there was not enough money for food. Participants selected the answer that best described them on a 5-point scale (1 = *almost never* to 5 = *almost always*) and an average was generated (α = 0.87). The scale showed adequate psychometric properties in the population it was originally developed, and these properties have since been replicated in different populations [35,36].

### 2.3. Procedure

The study protocol was approved by the university’s Institutional Review Board (IRB-20202-02-051) and consent was obtained. Participants attended the laboratory for a single testing session (approximately 30 min) in August–September 2020 (after easing of local lockdown for COVID-19). The tasks and survey questions were presented using Qualtrics and the SCPST was presented on the computer in high resolution. Participants initially completed current hunger measures, followed by the SPST and MPST (order randomly determined). Participants then completed the SCPST, provided user feedback and completed the TFEQ, food insecurity and demographic information questions. Participants also completed selected subscales from the Depression, Anxiety and Stress Scale ([37]; depression and anxiety), Body Attitudes Questionnaire ([38]; feeling fat, salience, disparagement and attractiveness) and the Physical Comparisons Scale-V3 [39] in full. Finally, participants also completed a researcher-created measure of satiety insecurity, however, these findings are not presented here. Participants provided their height and weight before having their height and weight measured by a researcher using a stadiometer and weight scales respectively. Participants were then debriefed and thanked for their time.

### 2.4. Statistical Analysis

Within our pre-registration we stated data exclusion criteria, but we did not state a confirmatory statistical plan due to the exploratory nature of the study. Variable scores for each participant were converted to z-scores and participants with z-scores over +/−3 were identified as outliers. Pearson’s correlation analysis (see Appendix A Appendix A) and linear regression analysis were used to assess the relationships between the three PSTs and the determinants of portion size. The Bland–Altman method for comparing two measures was used to determine if one (or both) of the simplified PSTs could be used as substitute for the SCPST [34]. Chi-square tests and t-tests were used to analyze the responses to the user feedback questions (see Appendix A section 3.0). Analysis was carried out using SPSS v26 (IBM). Self-report and researcher measured BMI was calculated with the following equation:weight (in kg)/(height^2^ (in m))(1)

## 3. Results

### 3.1. Participant Characteristics

Overall, 6.7% (*n* = 10) of participants reported they were currently dieting and the majority (90.7%, *n* = 136) described their diet as an omnivorous diet. Other diets reported included: vegetarian: 2.7% (*n* = 4); vegan: 0.7% (*n* = 1); and other: 6.0% (*n* = 9, e.g., diets in line with religious beliefs). Only 12 (8.0%) participants indicated they had engaged in exercise prior to completing the study and, subsequently, no further analysis was carried out. Average self-reported BMI (*M* = 22.06, *SD* = 3.73) was significantly correlated with researcher-measured BMI (*r* = 0.98, *p* < 0.001; *M* = 21.84, *SD* = 3.92). Descriptive statistics for participant characteristics and individual foods per PST and correlations between PSTs are reported in the Appendix A. Males and females selected similar-sized portions across all three versions of the PST (SCPST: *t*(147) = 0.72, *p* = 0.474; SPST: *t*(143) = −0.17, *p* = 0.865; MPST: *t*(147) = 0.25, *p* = 0.802). Average portion selections per task (in kilocalories) were: SCPST: *M* = 380.39 (*SD* = 114.82); SPST: *M* = 332.23 (*SD* = 96.37); MPST: *M* = 295.19 (*SD* = 101.63).

### 3.2. Regression Analysis

A series of regression analyses determined if the variables that predict SCPST also predict SPST and MPST. Current hunger, BMI, average liking, filling and familiarity, uncontrolled eating, cognitive restraint, emotional eating and food insecurity were entered as predictor variables and the three PSTs as outcomes in separate regression models.

SCPST. Overall model was significant, *F*(9118) = 4.78, *p* < 0.001, and explained 26.7% of variance. Average liking was the only significant predictor of SCPST, *B* = 4.92, *t*(118) = 5.66, *p* < 0.001.

SPST. Overall model was significant, *F*(9114) = 7.77, *p* < 0.001, and explained 38.0% of variance. Average liking (*B* = 4.69, *t*(114) = 6.98, *p* < 0.001), average familiarity (*B* = −1.18, *t*(114) = −2.04, *p* = 0.043), and cognitive restraint (*B* = −31.44, *t*(114) = −2.30, *p* = 0.023) were significant predictors of SPST.

MPST. Overall model was significant, *F*(9118) = 5.80, *p* < 0.001, and explained 30.7% of variance. Average liking (*B* = 4.60, *t*(118 = 6.13, *p* < 0.001) and cognitive restraint (*B* = −35.91, *t*(118) = −2.39, *p* = 0.018) were significant predictors of MPST.

### 3.3. Bland–Altman Agreement Analysis

A Bland–Altman analysis is used to examine the level of agreement between two measurements that aim to assess the same measure by graphically presenting where 95% of the differences between the two measures would be found [35] and the magnitude of any bias [36]. The difference between and average of the measures are calculated and examined using a one-sample t-test to check for significant differences from 0.

SPST/SCPST: Mean difference of 47.07 (*t*(145) = 8.67, *p* < 0.001, 95% CI (36.34–57.80)) and standard deviation of 65.58 (*SE* = 5.43); as the t-test revealed significant differences no further analysis was conducted.

MPST/SCPST: Mean difference of 85.20 (*t*(149) = 15.66, *p* < 0.001, 95% CI (72.78–93.79)) and standard deviation of 65.12 (*SE* = 5.62); as the t-test revealed significant differences no further analysis was conducted.

### 3.4. Generating an Equation to Estimate Portion Size

It is possible the lack of agreement between SCPST and SPST/MPST is due to the smaller range of images available and consequently, a smaller range of calorie portions (i.e., maximum calories for SPST/MPST were 740 kcal and SCPST was 1000 kcal). Therefore, we generated an equation to estimate the equivalent portion size that would be selected in the SCPST (SCPST_estimated_) from SPST or MPST. As there was no significant difference between the portion sizes selected by males and females, a single equation was generated. We conducted two regressions entering SPST and MPST as predictor variables alongside average liking (see Section 3.2.) and SCPST (SCPST_actual_) entered as the outcome variable. New variables were then created using the generated equations and additional Bland–Altman analyses conducted to assess the agreement between SCPST_actual_ and SCPST_estimated_ from SPST and MPST.

*SPST and average liking predicting SCPST_actual_.* Overall model was significant, *F*(2136) = 150.88, *p* < 0.001, and explained 68.9% of the variance. The following equation for SCPST_estimated_SPST_ was created:SCPST_estimated_SPST_ = 41.303 + (1.002 × SPST) + (0.089 × average liking)(2)

A one-sample t-test revealed a mean difference of −0.11 (*t*(138) = −0.02, *p* = 0.985, 95% CI [−11.08–10.87]) and *SD* = 65.432(*SE* = 5.55). The mean of the scores significantly predicted difference in scores (*F*(1137) = 15.38, *p* < 0.001) with a unstandardized *ß* = 0.20. The Bland–Altman plot and regression analysis (Figure 4) suggest the presence of proportional bias. As the predicted value on the *y*-axis is negative (−43.89) at the lowest value on the *x*-axis (164.96) and it is positive (68.73) at the highest value (722.46), this suggests that proportional bias is biphasic [36], such that at low SCPST levels, SCPST_estimate_SPST_ scores overestimate values and when SCPST levels are high, SCPST_estimate_SPST_ underestimates the values.

*MPST and average liking predicting SCPST_actual_.* Overall model was significant, *F*(2140) = 157.95, *p* < 0.001, and explained 69.3% of the variance. The following equation for SCPST_estimated_MPST_ was created:SCPST_estimated_MPST_ = 74.601 + (0.909 × MPST) + (0.619 × average liking)(3)

Figure 5 shows the comparison of MPST to SCPST_estimated_MPST_. A one-sample t-test revealed a mean difference of −0.07 (*t*(142) = −0.01, *p* = 0.990, 95% CI (−10.77–10.63)) and *SD* = 64.72 (*SE* = 5.41). The mean of the scores significantly predicted difference in scores (*F*(1141) = 15.58, *p* < 0.001) with a unstandardized *ß* = 0.20. The Bland–Altman plot and regression analysis (Figure 5) suggest the presence of proportional bias. The proportional bias is biphasic; the predicted value on the *y*-axis is negative (−40.11) at the lowest value on the *x*-axis (179.77) and is positive (67.19) at the highest value (718.93). At low SCPST levels, SCPST_estimate_MPST_ scores overestimate values and when SCPST levels are high, SCPST_estimate_MPST_ underestimates the values.

## 4. Discussion

Given the contribution of the ‘portion size effect’ to obesity [7], it is important that eating behavior researchers have useful, valid and easy to administer measures. Currently SCPSTs are regularly used but have numerous limitations (see Introduction). Therefore, we tested the properties of two simplified PSTs (slider version with five images (SPST)/multiple-choice version with seven images (MPST)) and assessed their agreement to the SCPST.

We replicated previously established associations between determinants of portion size and the simplified PSTs. For example, uncontrolled eating was significantly and positively associated with SPST and MPST (in line with Spence et al. [30]) and current hunger was significantly and positively associated with SPST (with a similar magnitude to Robinson et al., [9]). Average liking ratings of the foods were significantly and positively associated with, and predicted, all three versions of the PST and cognitive restraint was a significant predictor of SPST and MPST. In contrast to some studies (e.g., [9,40]), we did not find any significant differences in portion sizes selected between males and females. However, within our laboratory group, administering the standard PST on samples of Singaporeans has not consistently produced gender differences in selected portion sizes (see Appendix A section 5.0).

Results suggests the simplified PSTs could be used in lieu of the SCPST as a proxy for portion size. The simplified PSTs reduce researcher and participant burden and can be easily implemented into online studies to capture estimated portion sizes for a wide array for foods across large samples.

Whilst we show initial support for the use of simplified PSTs to capture estimated portion sizes, we do recommend findings are replicated in the laboratory using the SCPST or ad libitum feeding tasks. This is due to a lack of agreement between the average portion sizes as reported by the SPST/MPST and the SCPST as indicated by the Bland–Altman analysis. It is possible that the differences are due to the limited number of images presented in the simplified PSTs and in turn portion sizes available for participants to select. Consequently, we generated estimated SCPST values from the SPST/MPST and average liking. Results revealed that the values for average portion sizes measured by these approaches were not significantly different to the SCPST, therefore were in agreement, but did show proportional bias. This indicates that the simplified PSTs and the SCPST do not agree equally across all values, and we recommend researchers exert caution when drawing conclusions involving very small or very large portion sizes.

Our study does have some limitations. Firstly, the SCPST was always completed after the simplified PSTs which could have contributed to potential order effects. That said, several participants that provided free response feedback preferred the SCPST as it was more interactive. This could suggest that even though participants always completed it last, participants remained engaged when they completed it. Secondly, the study was conducted in a Singaporean sample involving target foods typical to the local context. Additionally, norms regarding portion size may vary between Singapore and other cultures (e.g., the United States, Europe). Notably, the simplified PSTs were limited to portion sizes of 20–740 kcal. Whilst this may be an appropriate range for the current sample, this range may be less suitable in cultures that have normalized larger portion sizes. Only one participant selected an average portion size over 740 kcal on the SCPST and less than 1.5% of participants have been found to report ideal portion sizes of over 740 kcal in multiple studies within our lab (see Appendix A section 5.0). In addition, the sample consisted largely of university students and, therefore, their selected portions may not reflect the average portion sizes of the wider population. Despite this, PSTs in general have been widely used outside of student samples including with children [22,41] and clinical populations [42]. Finally, the simplified PSTs presented here are developed as proxies for the SCPST and have not been compared with actual ad libitum food intake. Further studies are required to examine how well the SPST and MPST can be used as an estimate of food intake.

In addition, one potential caveat to the SPST is image size. In the current study, image size for the SPST was restricted to the parameters possible within the survey platform used (Qualtrics) and images had to fit along the continuum of the slider. As a result, the SPST images (120 × 120 px) were smaller than the MPST images (250 × 250 px) and we recommend that researchers use images in as large a format as possible to ensure changes are distinguishable whilst maintaining the overall presentation (i.e., all images run along the continuum) and to be wary that these may differ across platforms.

## 5. Conclusions

As increased amounts of psychological and behavioral research continues to move to an online platform, the creation of a valid online PST is important. We show that simplified versions of the SCPST can be used effectively online as an analogue of estimating ideal portion size, especially if used in combination with average liking of foods. Our findings suggest that the SPST and MPST can be predicted by determinants of portion size in line with previous patterns found with the SCPST. Our novel simplified PSTs can be incorporated easily into online research surveys and applied to a range of food items to test initial ideas prior to testing them in a laboratory setting.

## Figures and Tables

**Figure 1 foods-10-01121-f001:**
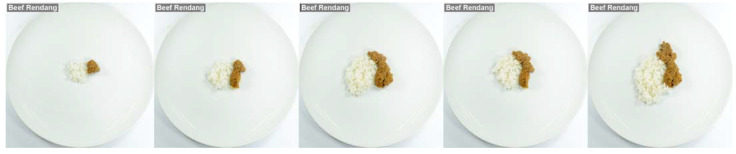
Sample images from the computerized portion selection task.

**Figure 2 foods-10-01121-f002:**
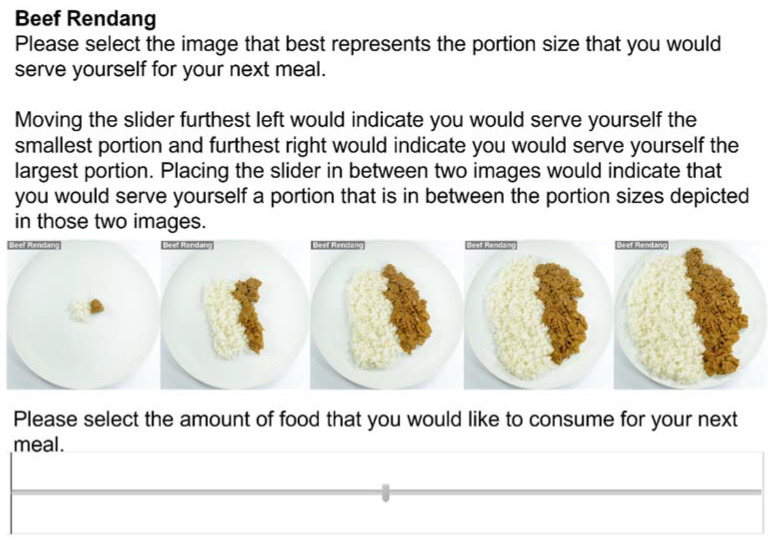
Example of the ‘slider’ version of the simplified portion size task (SPST).

**Figure 3 foods-10-01121-f003:**
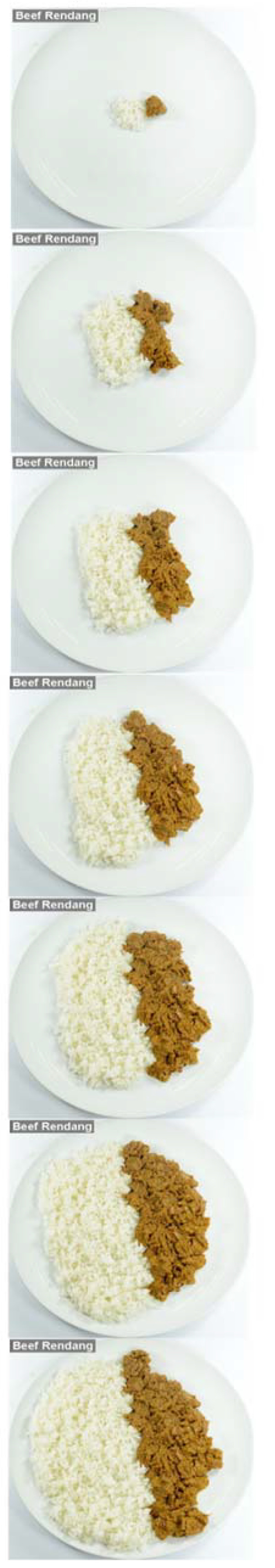
Example of the ‘multiple-choice’ version of the simplified portion size task (MPST).

**Figure 4 foods-10-01121-f004:**
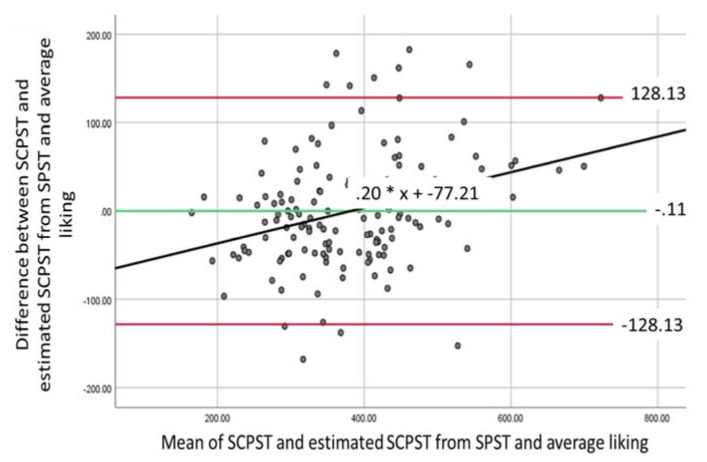
Bland–Altman plot for SCPST and SCPST_estimate_SPST._

**Figure 5 foods-10-01121-f005:**
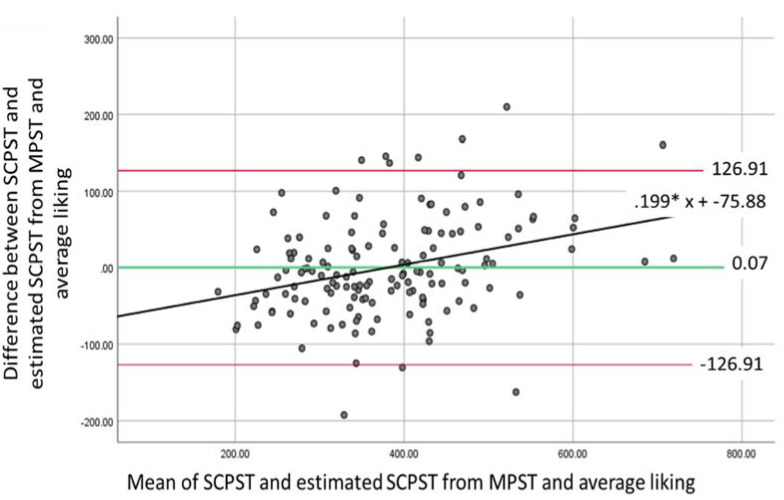
Bland–Altman plot for SCPST and SCPST_estimate_MPST._

**Table 1 foods-10-01121-t001:** Corresponding kcal portions presented as images to participants in the SPST and MPST.

	Kcal						
Task	Image 1	Image 2	Image 3	Image 4	Image 5	Image 6	Image 7
SPST	20	200	380	560	740	-	-
SCPST image ^+^	1	10	19	28	37	-	-
MPST	20	140	260	380	500	620	740
SCPST image ^+^	1	7	13	19	25	31	37

SPST = slider portion selection task, MPST = multiple-choice portion selection task, ^+^ = the image number in the standard computerized portion selection task (SCPST; e.g., 50 images depicting 20–1000 kcal portions).

## Data Availability

The data presented in this study are openly available in the Data Repository for Nanyang Technological University [DR-NTU] at https://doi.org/10.21979/N9/JVOO9A, created on 31 August 2020 and https://doi.org/10.21979/N9/1CYUKO, created on 31 August 2020. The study was preregistered (https://osf.io/vhzd2, created on 31 August 2020) and a copy of the questionnaire is available here. To request access to the stimuli please contact the corresponding author.

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
