# Peer review of "Development of a Simplified Portion Size Selection Task"

_foods, 2021, doi:10.3390/foods10051121_

Round 1

Reviewer 1 Report

Reviewer's Comments to Author:

Recommendation: Minor Revision

Manuscript Foods-1195463 with title "Development of a Simplified Portion Size Selection Task" provides an interesting study to develop portion size which is an important determinant of energy intake. Development and validation adequate tools for measuring portion size are still required because standard measures, such as ab-libitum designs, and currently available computerized portion selection tasks (PSTs) have several limitations.

I have the following comments:

Comments:

Abstract

  1. The abstract should state the study period, study type and sampling technique.
  2. Please clarify more the purpose of the study.
  3. Please revise your findings (results/numbers) and originality of the research.

Methodology

The manuscript shows as exploratory study using some paper-based questionnaire and pictures.

Please clarify all data collection

Please provide the sampling technique, inclusion criteria, and exclusion criteria

Whether is there Ethical permission for this study type

Questionnaire

Please provide the validity and reliability of used questionnaires

RESULTS and DISCUSSION:

Sociodemographic data are missing, at least presented collectively in the Appendix.  I think it would be useful to know: highest education level (elementary school, high school, higher education), place of residence (rural area, urban area, suburban area), marital status (unmarried, married/cohabitating, divorced, widow/widower), employment status (employed, unemployed, student, retired, student with a job),   engaging in physical exercise (never, sporadically, occasionally, moderately and intensively), time in hours spending watching TV or in front of the computer,  chronic diseases status (cardiovascular diseases, diabetes,  high cholesterol, arterial hypertension,  gastric disorders,    intestinal disorders)  food allergie/intolerance status (lactose intolerance,  casein,    gluten,  nuts,  shellfish)  whether they are following specific eating practices (raw foodism, frutarianism, vegetarianism, veganism, flexitarianism, caloric restriction, religion restrictions, other, general principles of healthy eating), whether  they are life experience an episode of any eating disorders (bulimia, anorexia, binge-eating) and whether they are responsible for food supplies (yes/no) or monthly income. Or indicate if you are excluded some of these respondents from study. Please define statistical methods in the footnote and the content of the tables or Figures should be such that the data are of sufficient resolution for comfortable reading.

Conclusions

Please write a clear conclusion. Revise your findings and originality of the research and should provide a clear scientific justification for the study.

Literature

The manuscript shows the relevant references related to this study.

Author Response

We thank Reviewer 1 for their time in reading our manuscript and providing helpful feedback. All changes to the manuscript have been made with track changes.

Abstract

  1. The abstract should state the study period, study type and sampling technique.
  2. Please clarify more the purpose of the study.
  3. Please revise your findings (results/numbers) and originality of the research.

Regarding suggestions #1 and #3 for the abstract, we have made this information clearer in the main text of the manuscript, instead of adding them into the abstract. Given the limited length of the abstract, these methodological details may take up too much space that is in competition with more important points to summarize about the study. Likewise, we have insufficient space within the abstract to report statistics and numeric values for results given the multiple results we summarize in the abstract. This information is clearly accessible to readers in section 3. Results.  Suggestion 1 has been reported in section 2.3 Procedure (line 226-227).

Our abstract has been revised to more clearly address the purpose of the study (Lines 13-15);

“The objective of the current study was to develop and test robust and valid measures of portion size that can be readily prepared by researchers and be reliably utilized for remote online data collection.”

Methodology

Please clarify all data collection

Please provide the sampling technique, inclusion criteria, and exclusion criteria.

We have updated our exclusion criteria, so readers are aware that we did not recruit participants who had a current or historical diagnosis of eating disorders (Line 115). Our other criteria (age) have already been stated (Line 114-115). Our sampling technique is reported in Lines 112-113. Participants were volunteers that were signed up to a research participation portal where university studies are posted. In addition, we have stated our Study period in the procedure section. Please let us know if any further clarification is required.

Whether is there Ethical permission for this study type

Yes, we stated in section 2.3. Procedure that ethical approval was provided by the University’s Institutional Review board (Lines 224-225). Please let us know whether this is suitable placement, or whether another section is more appropriate.

Questionnaire

Please provide the validity and reliability of used questionnaires

We have added further information to the Three Factor Eating Questionnaire (Lines 211-212) and the Food Insecurity measure (Lines 219-220) which highlight the validity and reliability of each measure. We provide citations for the reader to further find out about the validity and reliability of these measures.

RESULTS and DISCUSSION:

Sociodemographic data are missing, at least presented collectively in the Appendix.  I think it would be useful to know: highest education level (elementary school, high school, higher education), place of residence (rural area, urban area, suburban area), marital status (unmarried, married/cohabitating, divorced, widow/widower), employment status (employed, unemployed, student, retired, student with a job),   engaging in physical exercise (never, sporadically, occasionally, moderately and intensively), time in hours spending watching TV or in front of the computer,  chronic diseases status (cardiovascular diseases, diabetes,  high cholesterol, arterial hypertension,  gastric disorders,    intestinal disorders)  food allergie/intolerance status (lactose intolerance,  casein,    gluten,  nuts,  shellfish)  whether they are following specific eating practices (raw foodism, frutarianism, vegetarianism, veganism, flexitarianism, caloric restriction, religion restrictions, other, general principles of healthy eating), whether  they are life experience an episode of any eating disorders (bulimia, anorexia, binge-eating) and whether they are responsible for food supplies (yes/no) or monthly income. Or indicate if you are excluded some of these respondents from study. Please define statistical methods in the footnote and the content of the tables or Figures should be such that the data are of sufficient resolution for comfortable reading.

Of the available sociodemographic information we collected, we have reported marital status, highest level of education and reported household income in section 2.1. Participants. For ease of readers that may be from different countries and use different education systems we emphasise that all participants were university students.  Current dieting status and dieting category (e.g. omnivore, vegetarian etc.) is reported in section 3.1. Participant Characteristics. The information on the lines of the figures have been made clearer, please let us know if further adjustments are needed for the figures.

Conclusions

Please write a clear conclusion. Revise your findings and originality of the research and should provide a clear scientific justification for the study.

We have separated our final paragraph of the discussion (Section 4) to create a new section (5. Conclusion). Here we have stated why the study was conducted (a greater move to online research and a valid portion size selection task is needed),and have highlighted our results (our simplified PST can be incorporated into online research to test initial research ideas).

Reviewer 2 Report

Further little qualitative description of results would be fine for a broader audience, less involved in econometrics.

Author Response

Reviewer 2.

We thank Reviewer 2 for their time in reading our manuscript and providing helpful feedback. All changes to the manuscript have been made with track changes.

Further little qualitative description of results would be fine for a broader audience, less involved in econometrics. 

The overall goal of our study was to validate and test the psychometric properties of two simplified portion size selection tasks and compare them to the standard computerized task commonly used which has required use to use and report several quantitative analyses. Given feedback from Reviewer 3 we have now also added in some additional qualitative findings into the Supplementary Materials that cover the free response feedback participants made. Please let us know if this is sufficient or if further clarification is needed.

Reviewer 3 Report

This is a good paper. The authors do a good job explaining the objective, methods and results. Overall, this paper is of high quality. I only have a couple minor comments (listed below).

Line 111- How the people become involved in the research participation system? Could this be a limitation of this study?

Line 120- How was the researcher measured BMI determined?

Line 199- Were the results of the open comment box analyzed?

Line 351- Since the participants completed the SCPST after the PSTs, do the authors think fatigue or boredom was a factor in the results?

Author Response

We thank Reviewer 3 for their time in reading our manuscript and providing helpful feedback.  All changes to the manuscript have been made with track changes.

Line 111- How the people become involved in the research participation system? Could this be a limitation of this study?

We have added a footnote into section 2.1. Participants to further explain the research participation system. This system is often used for psychological research within university settings. We have also added a comment into the limitations paragraph of section 4. Discussion. Here we express that this could be a limitation of the study and further studies are required but we also use this opportunity to highlight the wide array of populations PSTs have been used previously.  

Line 120- How was the researcher measured BMI determined?

We have moved BMI descriptives to section 3.1. Participant Characteristics so that the methods of researcher measured BMI are presented before. In section 2.3 Procedure we had added in the equipment and methods for capturing height and weight used to calculate BMI in our study. We have also added to section 2.4. Statistical Analysis to highlight we used the standard equation for calculating BMI.

Line 199- Were the results of the open comment box analyzed?

No, all participants used the opportunity to provide comments in the open box. However, we have now added a small section into the Supplementary Materials that summarises the written feedback received.

Line 351- Since the participants completed the SCPST after the PSTs, do the authors think fatigue or boredom was a factor in the results?

We have acknowledged that it possible our study may have been susceptible to order effects in section 4. Discussion (Lines 367-370) and whilst we are unable to further elaborate on the effect this may have had on our findings we had included some additional information taken from our free response feedback which may suggest participants remained engaged (see supplementary materials).